# Moral Breakdowns and Ethical Dilemmas of Perioperative Nurses during COVID-19: COREQ-Compliant Study

**DOI:** 10.3390/healthcare11131937

**Published:** 2023-07-04

**Authors:** Amalia Sillero Sillero, Raquel Ayuso Margañon, Maria Gil Poisa, Neus Buil, Eva Padrosa, Esther Insa Calderón, Elena Marques-Sule, Carlota Alcover Van de Walle

**Affiliations:** 1ESIMar (Mar Nursing School), Parc de Salut Mar, Universitat Pompeu Fabra Affiliated, 08003 Barcelona, Spain; asillero@esimar.edu.es (A.S.S.); mgilp@esimar.edu.es (M.G.P.); epadrosas@esimar.edu.es (E.P.); einsa@esimar.edu.es (E.I.C.); calcoverv@esimar.edu.es (C.A.V.d.W.); 2SDHEd (Social Determinants and Health Education Research Group), IMIM (Hospital del Mar Medical Research Institute), 08003 Barcelona, Spain; 3Nursing Care Research, IIBSANT PAU, Hospital Santa Creu i Sant Pau, 08025 Barcelona, Spain; neus@diazbuil.com; 4Physiotherapy in Motion, Multispeciality Research Group (PTinMOTION), Faculty of Physiotherapy, Department of Physiotherapy, University of Valencia, 46010 Valencia, Spain; elena.marques@uv.es

**Keywords:** COVID-19 pandemic, ethical dilemmas, moral dilemmas, nurses, nursing, qualitative study

## Abstract

(1) Background: The COVID-19 pandemic has led to an increase in the complexity of caregiving, resulting in challenging situations for perioperative nurses. These situations have prompted nurses to assess their personal and professional lives. The aim of this study was to explore the experiences of perioperative nurses during the first wave of the COVID-19 pandemic, with a specific focus on analyzing moral breakdowns and ethical dilemmas triggered by this situation. (2) Methods: A qualitative design guided by a hermeneutical approach was employed. Semi-structured interviews were conducted with 24 perioperative nurses. The interviews were transcribed and thematically analysed following the Consolidated Criteria for Reporting Qualitative Research (COREQ) guidelines. (3) Results: The findings revealed three main categories and ten subcategories. These categories included the context in which moral breakdowns emerged, the ethical dilemmas triggered by these breakdowns, and the consequences of facing these dilemmas. (4) Conclusions: During the first wave of COVID-19, perioperative nurses encountered moral and ethical challenges, referred to as moral breakdowns, in critical settings. These challenges presented significant obstacles and negatively impacted professional responsibility and well-being. Future studies should focus on identifying ethical dilemmas during critical periods and developing strategies to enhance collaboration among colleagues and provide comprehensive support.

## 1. Introduction

Nurses frequently encounter moral and ethical decisions inherent in their clinical practice [1,2]. Several literature reviews highlight the importance of developing a comprehensive understanding of the nursing profession’s ethical dilemmas to enhance ethical competence [3,4,5]. The COVID-19 pandemic, though, introduced a new array of challenging and intricate ethical situations and discussions. For example, nurses have faced the ethical dilemma of caring for seriously ill patients who are in end-of-life situations and separated from their families [6,7]. These unprecedented circumstances have heightened the complexity of ethical decision-making and raised novel ethical concerns for healthcare professionals.

Despite nurses’ awareness of professional, ethical principles, applying these principles during the pandemic has posed significant challenges due to a need for more guidance and support from management [8]. As a result, nurses have experienced moral and emotional distress [9,10], including feelings of grief, anger, frustration, compromised moral integrity, and diminished self-esteem [11,12,13]. Professionally, they have encountered difficulties in interpersonal communication, lack of leadership, and desire to leave the profession [13,14,15]. 

The experiences of perioperative nurses during the pandemic are particularly intriguing. In addition to the challenges mentioned earlier, they have had to adapt to significant structural changes in surgical units, such as their conversion into COVID-19 care units [16,17,18] or being reassigned to intensive care units (ICUs) [19]. There is scarce available evidence regarding the specific experiences of perioperative nurses in these environments and, more importantly, how they may contribute to various ethical dilemmas and develop potential moral breakdowns among nurses.

While ethics and morality are often used interchangeably, they possess distinct nuances. This study will adopt Zigon’s theoretical framework [20] to explore the ethical dimensions of perioperative nursing experiences during the COVID-19 pandemic. At the same time, adopting the Evidence-Based Nursing Model provides a comprehensive approach to guide nursing practice in high-quality care.

According to Zigon, morality relates to non-reflexive norms and awareness of ethical dilemmas, encompassing freedom and moral choice. On the other hand, ethics is associated with cultivating virtues as part of personal growth, achieved through practice and engagement in specific activities [21], such as the professional realm. Significantly, both concepts coexist in the daily lives of nurses [22] as everyday life is not a static entity but rather a way of being immersed in tasks and activities anchored in the present moment [23]. In this state, individuals may lose self-perception, prioritizing the needs of others, accepting the norm of existence, and losing sight of their ultimate reality.

Nonetheless, Zigon asserts that certain events or situations disrupt this state and compel individuals to assess and consciously reflect on the most appropriate response. These instances, which prompt individuals to revisit themselves and challenge dominant conceptions, norms, or beliefs, are what Zigon terms “moral breakdowns”, marking the threshold at which he distinguishes morality from ethics [22,24]. Thus, morality represents an unconscious way of being shaped by embodied moral dispositions, beliefs, conceptions, hopes, expectations, and more [24]. Simultaneously, ethics corresponds to conscious questioning and initiating an ethical work process aimed at assimilating new provisions and returning to moral action in the social world. A moral breakdown propels individuals to engage in self-work, leading them to what Zigon refers to as an ethical moment, where possibilities unfold, enabling conscious choices [20].

Understanding the numerous ethical challenges that nurses encountered during the pandemic is essential for a comprehensive understanding of the profound and extensive impact of the pandemic on the nursing profession. This understanding is crucial in informing and enhancing the continuous emotional, psychological, and practice support provided to nurses during and even after the pandemic.

Drawing upon Zigon’s theoretical framework of ethics and morality, the primary objective of this study is to delve into perioperative nurses’ experiences during the initial wave of the COVID-19 pandemic. Our focus is directed explicitly towards investigating potential moral breakdowns and ethical dilemmas in this unusual situation. By conducting this exploration, we aim to foster a more profound comprehension of the distinctive ethical challenges perioperative nurses face in a global health crisis. Ultimately, the insights gained from this study will inform the development of strategies to support and enrich perioperative nurses’ well-being and professional practice. 

## 2. Materials and Methods

### 2.1. Study Design and Participants

A qualitative design guided by a hermeneutical approach [25] was used to explore the experiences of perioperative nurses during the first wave of the COVID-19 pandemic through an analysis of their narratives. Participants were recruited on-site in a university hospital in Barcelona, Spain, from June to July 2020. Potential participants were informed of the purpose of the study and were invited to participate. The inclusion criterion was to have worked as a perioperative nurse during the first wave of the COVID-19 pandemic (14 March–1 June 2020). A total of 24 perioperative nurses were interviewed until data saturation was reached when interviews did not offer new information [26]. Interventional studies involving animals or humans, and other studies that require ethical approval, must list the authority that provided approval and the corresponding ethical approval code.

### 2.2. Data Collection

Face-to-face semi-structured interviews were conducted from June to July 2020. Prior to the start of the interview, participants were informed again about the study and the implications of participating, and they were given a written consent form. During the interview, a note-taker was present to keep track of the topics discussed. Interviews were audio-recorded for subsequent literal transcription. Interviews were transcribed and pseudo-anonymized. Interviews lasted between 40 and 60 min and covered aspects related to nurses’ experiences during the first wave of the pandemic, including topics such as their thoughts and feelings about treating COVID-19 patients or being assigned to different units, coping mechanisms, and recommendations for future solutions in public health crises. Background information was also collected, including gender, age, household composition, years of experience, working shift, type of contract, and if they had been sent to the reserve, reassigned to other units, treated COVID-19 patients, or had a COVID-19-related sick leave.

### 2.3. Data Analysis

An interpretative hermeneutical approach was used, following Crist and Tanner’s recommendations [25]. An abductive thematic analysis strategy was divided into five stages: identification of deductive categories and subcategories, familiarization with the data, identification of new categories and subcategories, review of deductive and inductive categories and subcategories, and naming of final categories and subcategories.

We started preliminary data analysis right after the first interviews were conducted to assess whether and when informational saturation was reached, and pre-defined deductive codes for thematic analysis were determined. After transcribing and pseudo-anonymizing all the interviews, three analysts independently coded the same randomly chosen transcripts. Forty-three data codes were found, distributed in ten code groups. Then, the three researchers met to discuss and resolve discrepancies related to the use of deductive categories and subcategories, as well as the generation of additional themes. These additional themes were either placed as subcategories into a deductive category or grouped as new categories. Once a category tree was agreed on, three researchers verified the coherence of categories and subcategories and checked for clear distinctions between them. Thereafter, three researchers systematically coded the rest of the transcripts and met weekly to refine categories and subcategories as the coding process moved forward [27]. The results of data analysis were discussed with the full research team to facilitate the interpretation and discussion of findings. All study authors hold a PhD degree and have extensive experience in qualitative research, including qualitative data analysis. Their expertise and qualifications enable them to perform the coding and analysis of the transcripts effectively. Analysis was conducted using ATLAS.tiWindow Software (23.0.8.0).

To ensure rigour throughout the research process, we followed multiple steps adapted from Lincoln and Guba’s criteria of evaluation (1985) [28]. Credibility, transferability, confirmability, and trustworthiness were guaranteed by integrating researcher triangulation, considering reflexivity in every stage of the research project, undertaking peer debriefing after each interview, managing data systematically, and exploring potential interpretations of findings. Peer checking of team members was also applied, along with participants’ feedback. The Consolidated Criteria for Reporting Qualitative Research (COREQ) were followed [29]. See the Checklist in the Appendix A for comprehensive information.

### 2.4. Ethical Considerations

Prior to implementation, the study was guaranteed ethics approval by an Official Ethics Review Board (IIBSP-COV-2020-58). Beyond that, we followed the standards established in the Declaration of Helsinki (2013): participants had access to an information sheet of the study and signed a written consent form prior to data collection [30]. Data were treated according to Regulation (EU) 2016/679 of the European Parliament and the Council of 27 April 2016 on the protection of natural persons with regard to the processing of personal data and on the free movement of such data and repealing Directive 95/46/EC (General Data Protection Regulation) [31].

## 3. Results

Twenty-four nurses met the inclusion criteria, completed the study, and were included in the analyses. All nurses were women, their ages ranged from 28 to 60 years old, and they had been working as nurses between 3 and 40 years, the mean being 20 years of experience. Most of them worked day shifts (n = 17) and had permanent jobs (n = 20). During the first wave of the pandemic, the majority were sent to the reserve at some point (n = 17), meaning that they had to stay at home until they were called to the frontline, and half of them were reassigned to other units (n = 12). All of them but one treated COVID-19 patients. However, only 10 individuals went on sick leave due to COVID-19 during that period. The characteristics of the sample are presented in Table 1.

Three main themes were identified from the analysed narratives: (1) the context of moral breakdowns, (2) the ethical dilemmas triggered by moral breakdowns, and (3) the consequences of facing ethical dilemmas. The categories and subcategories are presented in Table 2. 

### 3.1. Context in Which Moral Breakdowns Emerged

According to the narratives of perioperative nurses, the feeling of being in a collapse situation was key to their experience during the first wave of the pandemic. They reported experiencing quick, poorly informed, and disorganised changes in their units and, by extension, in their tasks and activities. For instance, some of them were called to the frontline to take care of COVID-19 patients, reassigned to other units, asked to adopt new professional roles and tasks, or challenged with multiple schedule changes. And they were not necessarily trained for these changes (Q1,2). 

To add to this tension, they had to work under poor working conditions, dealing with a shortage of beds, equipment, medication, and professionals. In the interviews, it was salient that transforming resuscitation units into ICUs generated structural barriers to using the available resources, which impeded delivering patient care (Q3,4).

For many, this resulted in an emotional debacle. Nurses narrated how the situation became emotionally unbearable when they thought about their lack of knowledge to treat COVID-19 patients (Q5,6).

### 3.2. Ethical Dilemmas Triggered by Moral Breakdowns

All these factors combined were the perfect storm for moral breakdowns to burst. The interviews highlighted that perioperative nurses started to doubt the morality of how their profession was being performed due to the above-described context, and this triggered ethical moments in which they questioned not only their daily tasks and activities but also those of their peers and supervisors. 

The first ethical dilemma was the quandary between professional responsibility versus the incapacity to provide quality care. Nurses reported not having time or resources to take care of patients in a holistic manner. Consequently, some interviewees felt they were hindering the principle of justice, as they could not treat patients equally. Moreover, perioperative nurses described how they did not feel prepared or trained to treat critical patients, which interfered with their sense of professional duty (Q7,8). 

Also related to the morality of the nursing profession, nurses experienced ethical dilemmas when it came to respecting the right of patients to be accompanied by their families versus the restriction of visits in health institutions. These restrictions placed nurses in a situation in which they had to emotionally support patients knowing that what would help them the most would be to see their families, infringing the principles of autonomy and, as they wondered, even of beneficence (Q9). 

A third dilemma perioperative nurses referred to was the choice between patient care versus self-care. As depicted in the interviews, nurses were confronted with wanting to care for COVID-19 patients and not wanting to become infected and infect their families. This was compounded by the fact that most nurses were not adequately provided with PPEs, as described by the quotes referred on Table 3 (Q10,11).

Finally, breakdowns triggered by the pandemic opened the door to questioning the morality of peers, supervisors, and other health professionals. It is worth noting that these dilemmas might hold significant importance for both the informants and researchers. This is primarily due to their direct relevance in professional settings and the potential impact they can have on workplace relationships. Perioperative nurses that were sent to the frontline felt disappointed and sometimes distrusted the commitment to the profession of, on the one hand, colleagues that were sent to the reserve (Q12,13) and, on the other hand, supervisors who made those decisions (Q14). They also described feelings of skepticism related to the recruitment of nursing students while keeping experienced professionals in reserve (Q15). In that line, nurses that were sent to the reserve or kept their role in the perioperative service experienced guilt and regretted not being able to be there with their colleagues and patients (Q16) should the situation not have been dangerous for their families (Q17). 

### 3.3. Consequences of Facing Ethical Dilemmas

The ethical dilemmas mentioned above entailed multiple consequences, being emotional reactions the most remarked on in the interviews (Q18,19). This topic emerged in the narratives of perioperative nurses on how they experienced feelings of despair, powerlessness, fear, and rage; these sometimes triggered emotional and mental health effects, including anxiety, depression, or sleep disorders. 

Occasionally, facing these ethical dilemmas also led nurses to reconsider their profession, albeit with diverging outcomes. While some perioperative nurses questioned whether the profession was worth it (Q20), others re-awoke the significance of being a nurse and even considered changing from the surgical ward to other units (Q21).

Similarly, nurses narrated that working under such conditions strengthened interpersonal relationships within teams and identified interprofessional teamwork as one of the most helpful strategies to cope with difficulties (Q22), along with resilience (Q23,24). However, when it came to supervisors, nurses exposed multiple needs that had not been adequately addressed. In their narratives, they talked about feeling unsupported after their return, as if supervisors did not care about the emotional discomfort that the overall process of getting back to the surgical ward was causing them (Q25).

For many of the interviewed nurses, acknowledgement and support from their supervisors would have been key, not only for a safe return (Q26) but also for the management of future public health crises, coupled with better-articulated information flows and continuous training programs (Q27). 

## 4. Discussion

This study aimed to explore the experiences of perioperative nurses during the initial wave of the COVID-19 pandemic, focusing on moral breakdowns and the subsequent ethical dilemmas they faced. The constant conflict between providing care under adverse conditions and maintaining quality care posed significant challenges, leading to moral and emotional distress among nurses [32]. This study is one of the first of its kind to examine the morality of the nursing profession within the context of a pandemic, specifically from the perspective of nurses who rapidly adapted to new roles, tasks, and activities.

The discussion highlights various ethical principles that were at stake during the pandemic, aligning with challenges identified in recent nursing studies. These principles include autonomy, justice, respect for patients and their families, caregiving quality, safety in the working environment, and health of both nurses and patients [33,34,35,36,37,38,39,40].

The COVID-19 pandemic brought about unprecedented changes in healthcare, including strict physical distancing measures. Overnight, healthcare professionals found themselves dealing with biosecurity imperatives such as locked doors, restricted areas, and visitor restrictions [7]. While the interviewed nurses acknowledged the importance of these measures in preventing COVID-19 infections and the severity of the disease, they disagreed with the extent of visitor restrictions and the resulting isolation of patients from their families [41], they disagreed with the extent to which visitor restrictions and patients died alone [33,36]. Nurses became intermediaries between patients and their families, adding to the emotional burden they carried [36,42]. This challenging situation not only impacted the emotional well-being of nurses but also generated ethical and moral conflicts, as they questioned the imposed measures [11,12,36,43,44,45,46].

During the pandemic, nurses faced the dilemma of balancing their professional responsibilities and values. The principles of nonmaleficence (do no harm) and beneficence (doing good) were in tension, as nurses had to prioritize patient safety despite their own well-being [36,47,48]. The core duty of nursing practice is to provide care, restore health, alleviate suffering, and respect the rights of every patient [49]. However, in this exceptional situation, nurses found themselves torn between their caregiving duty to patients and the need to prioritize self-care. It is important to note that nurses’ duty of caregiving is not absolute and can become a dilemma when it conflicts with their personal beliefs [32]. According to the International Code of Ethics for Nurses [49], nurses also have a commitment to promote their own health and safety. In this study, perioperative nurses sought to strike a balance between their responsibilities to patients and their own rights, all within the unique circumstances of the pandemic. 

Perhaps the most disruptive ethical issues were the conflicts between the right to decide (the principle of autonomy) and patient safety, as well as the conflicts between the principles of beneficence and nonmaleficence [36,47,50]. Nurses faced ethical questions regarding their stressful and exhausting working conditions, such as the lack of protective equipment, limited knowledge and experience with COVID-19, and scarce protocols for the pandemic [36,51,52]. In this context, several questions arose, such as how to protect the principle of autonomy, to what extent medical committees should weigh patients’ rights against the working conditions of healthcare workers, and how nurses can ensure their own and their family’s safety [8,39,40,53]. One of the key solutions to these dilemmas lies in creating safe working conditions, which is essential for providing safe care to patients and preventing moral distress or harm among nurses [54,55,56]. We acknowledge that contextual factors and evolving circumstances can significantly influence the experiences and ethical challenges faced by perioperative nurses.

The principle of justice also played a role in the ethical challenges faced by nurses. Nursing leaders had to divide nursing personnel into different teams, such as first-line and reserve nurses, to reduce the risk of contagion. This change in professional roles caused feelings of unfairness among nurses, as some perceived they had assumed more risk than others [17,18]. Nurse leaders must apply the principle of justice by ensuring fair distribution of burdens and implementing adequate rotation among nurses to balance assignments and workloads, thereby reducing negative feelings [57,58,59,60].

Nurses relied on teamwork and peer support resilience to navigate these moral breakdowns. Nurse managers and leaders should continue to promote teamwork, recognize and reward strong teams, and foster an ethical work environment [47]. Resilience is crucial in managing ethical challenges, and nurses need to learn to adapt to new situations as a team [61,62]. Some participants in the study found these experiences to be opportunities for personal and professional growth, while others contemplated leaving the profession. Prolonged exposure to moral breakdowns can lead to compassion fatigue, highlighting the need for professional resilience and strategies to cope with stressful environments [63,64].

The studies by Jia et al. (2021) and Hossain and Clatty (2021) offer valuable insights into the ethical challenges and self-care strategies among nurses during the pandemic, albeit not specifically focused on perioperative nursing. Nonetheless, the findings from these studies can still provide relevant perspectives and contribute to our understanding of the broader ethical landscape in healthcare settings during this unprecedented time [42,64].

This pandemic presents an opportunity to learn and improve healthcare practices by establishing evidence-based strategies. Nursing managers should prioritize an ethical work environment and implement strategies to enhance nurses’ well-being, morale, and personal and professional development [65]. Ethics education and training should be provided to create greater awareness of ethical considerations and promote the application of professional values in daily practice [66].

### 4.1. Limitations and Strengths

This study acknowledges several limitations that should be highlighted. Firstly, the research sample was obtained from a single hospital in Spain, limiting the findings’ generalizability. Caution should be exercised when applying the results to a broader population.

Secondly, the study was conducted shortly after the peak of COVID-19 infections, and intense personal and professional emotional experiences may have influenced the participants’ responses during that time. Although the nurses had resumed their regular practice, some of them might still be in the process of reflecting on and comprehending their experiences.

Additionally, the study solely employed qualitative methods, and including quantitative research could have enriched the obtained data. Future studies could consider adopting a mixed-method approach combining quantitative and qualitative components to provide a more comprehensive understanding of the subject.

Despite the mentioned limitations, an essential strength of this study is its distinction as one of the few studies that have explored perioperative nurses’ experiences during the initial wave of COVID-19.

### 4.2. Implications of Findings

The findings of this study have significant implications for nurses and healthcare professionals. By being aware of the moral breakdowns and ethical dilemmas that may arise in the clinical environment during critical periods like the COVID-19 pandemic, nurses can better understand their ethical role as healthcare providers and maintain an ethical perspective in their daily clinical practice. It is crucial to prioritize ethics education and training in workplaces, and further research should assess how these moral breakdowns and ethical dilemmas impact the quality of patient care. Future studies should focus on identifying moral breakdowns and ethical dilemmas experienced by nurses during critical periods, with the ultimate goal of improving the overall well-being of healthcare professionals.

## 5. Conclusions

In this study, we investigated how perioperative nurses in the first wave of COVID-19 coped with, perceived, and were impacted by moral and ethical challenges, commonly referred to as moral breakdowns, which are often inevitable and prevalent in critical settings. These challenges posed significant hurdles that had to be addressed. It became crucial to prioritize the well-being of nurses on both personal and professional levels while simultaneously striving to enhance patient care. The participants in the study relied on their own resourcefulness to navigate the complexities of moral and ethical dilemmas.

The study revealed that the nature and duration of the response, as well as the severity, frequency, and duration of morally challenging situations, were perceived as having a detrimental effect on both professional responsibility and well-being. Crucially, the support that proved most beneficial was primarily centered around team support, emphasizing the importance of collaboration and solidarity among colleagues. Furthermore, management support after deployment was considered valuable, particularly when it demonstrated a comprehensive understanding of the working conditions and the shared experiences of the nursing staff.

Moreover, being well-informed and adequately prepared to confront diverse moral and ethical challenges within a demanding work environment can significantly contribute to stress management and the prevention of moral breakdowns. It is important to acknowledge that these problematic experiences, while exacting a personal toll, also engendered positive outcomes such as personal and professional growth and a transformed worldview.

The findings of this study have implications for organizations in developing support structures for nurses and other healthcare professionals, both before, during, and after their deployment. By recognizing the unique challenges faced by nurses and providing targeted support, healthcare organizations can foster an environment that promotes the well-being and resilience of their workforce. This study serves as a valuable resource to inform and shape organizational strategies and interventions aimed at supporting nurses and healthcare professionals in navigating moral and ethical challenges throughout their careers.

## Figures and Tables

**Table 1 healthcare-11-01937-t001:** Characteristics of the sample.

	Total N = 24
**Gender**	
Woman, frequency (%)	24 (100)
**Age,** mean (SD)	42.21 (9.80)
**Household composition**, frequency (%)	
Living alone	6 (25)
Living with children	11 (45.83)
Living with elder	2 (8.33)
Other	5 (20.83)
**Years of experience**, mean(SD)	19.83 (9.80)
**Working shift**, frequency (%)	
Day shift	17 (70.83)
Afternoon shift	4 (16.67)
Night shift	2 (8.33)
Other	1 (4.17)
**Type of contract**, frequency (%)	
Permanent	20 (83.33)
Temporary	4 (16.67)
**Sent to the reserve**, frequency (%)	17 (70.83)
**Reassigned to other units**, frequency (%)	12 (50)
**Treated COVID-19 patients**, frequency (%)	23 (95.83)
**COVID-19-related sick leave**, frequency (%)	10 (41.67)

All data are expressed as mean(SD) or frequency(percentage), as appropriate. %: percentage; SD: Standard Deviation.

**Table 2 healthcare-11-01937-t002:** Categories and subcategories of narratives encountered by perioperative nurses.

Categories	Subcategories
**1. Context in which moral breakdowns emerged**	*Quick and continuous changes*; *Working conditions, lack of material and human resources*; *Lack of knowledge to treat COVID-19 patients*.
**2. Ethical dilemmas triggered by moral breakdowns**	*Professional responsibility versus the incapacity to provide quality care*; *Respecting the right of patients to be accompanied by their families versus the restriction of visits in health institutions*. *Patient care versus self-care*; *Ethical dilemmas with peers and supervisors*.
**3. The consequences of facing ethical dilemmas**	*Physical, mental, and emotional health*; *Re-Consider the profession: find the meaning or abandon*; *Strengthened interpersonal relationships within teams*; *Resilience*; *Key factors for a safe return*.

**Table 3 healthcare-11-01937-t003:** Results table and participants’ quotes.

Quote	Category	Subcategory		Participant
Q1	Context in which moral breakdowns emerged	Quick and continuous changes	It was very chaotic at the beginning because, when we went from resuscitation to the ICU with intubated patients, it was a very uncontrolled change [...] the organization was deficient; we had material that we didn’t know if it was contaminated or not, we didn’t know which circuits we had to do, how we had to dress…	P1, 30y
Q2			Physicians, every day, it was a different team [...] every day a different treatment, because if a different team comes, then they tell you to pronate him the patient] less hours, or to give him this other medication. And the one [team] that comes the next day says a different thing, and this generated a lot of uncertainty	P6, 46y
Q3		Working conditions, lack of material and human resources	The schedule… I think it’s very hard what we did. 3 days in a row and 1 resting day. Many hours in a row, with goggles, gowns and masks, sweating and not going to the restroom, not even drinking a juice… You couldn’t bear it	P9, 28y
Q4			There was no space to work, lucky if we had respirators [...] there was a box, I don’t remember which one, that if you wanted to plug in the oxygen intake, you had to unplug the respirator because there weren’t enough [plugs], you know?	P1, 30y
Q5		Lack of knowledge to treat COVID-19 patients	I felt bad, bad, bad. Powerless. Because you don’t know what to do… There was no time to take care of them, I didn’t know… I couldn’t… What else can I do? I wanted to cry	P17, 41y
Q6			[...] I am an absolute zero, I don’t know how to deal with the patient… Fear, a lot of fear. And getting out of the reanimation area… My goodness, I was very nervous	P16, 52y
Q7	Ethical dilemmas triggered by moral breakdowns	Professional responsibility versus the incapacity to provide quality care	What we did in the resuscitation unit… I wouldn’t do it again. First of all, because of the patients because, I mean, being assigned there without professional experience and under those conditions… almost 10 hours wearing PPEs [Personal Protective Equipment]	P9, 28y
Q8			The way to intubate is totally different, for example with the gliescope, I had never seen it being done like this… a lot of intubations. And also there were only two [gliescopes]... for so many critical patients. Things like that… that everyone had to be intubated and two [gliescopes] for so many patients… I don’t know	P6, 51y
Q9		Respecting the right of patients to be accompanied by their families versus restriction of visits in health institutions	It’s just that I don’t understand, I don’t understand. Maybe build the shifts saying… even patients and family members come in half an hour, from this time to that time. Patients in odd rooms come at half past… some in the morning and some in the afternoon… right? But half an hour each, and patients in end-of-life phases, well, you give them PPEs and… let them be by their side. And that’s what I was asking myself and, I don’t know, I was questioning a lot of these things	P4, 31y
Q10		Patient care versus self-care	“[...] the fear that you didn’t know how to protect yourself either, because it was at the very beginning, and you didn’t have new PPEs everyday[...]. Then, my husband, who has heart disease. I called primary care, he is a doctor, and they sent him back home immediately, home quickly. [...] When I got home, of course, I was very afraid [...]. It was two types of fear: fear of the job, which I wasn’t familiar with, and fear of the disease, of being infected and infecting those at home”	P14, 60y
Q11			“Come to work and take risks, because we are not protected… because every day our colleagues were falling. That was a strange feeling [...] that in the end you were waiting to see when you fall, when my turn comes”	P18, 38y
Q12		Ethical dilemmas with peers and supervisors	“I think that this is a new war, and we are here, and this is our job, not staying at home. This is what I’m most sorry for, people who stayed at home and didn’t… who weren’t, didn’t come”	P16, 52y
Q13			I feel sorry, I feel sorry… It is not okay that not everyone participated in this”	P7, 46y
Q14			I received a WhatsApp message from a supervisor saying: tomorrow you are going to the operating room. I thought, I thought… uh, I thought inside of me: how delicate… after being sick for 22 days. A call, a “hello, how do you feel? How are you? We’ve thought… We’ve counted on you for tomorrow… to work. How are you? Are you okay to come to work tomorrow?” Knowing that there were reserve people at home, and that they hadn’t been sick, right?	P11, 33y
Q15			I don’t fully grasp why nursing students were there… While there were fellow nurses on the reserve, they were at home…	P22, 59y
Q16			And I had a bad time because I felt bad about not being able to help. And when I call the supervisors, they tell you that you can’t, that you are in the reserve because you need to be in your working area, which is the surgical ward	P19, 43y
Q17			If it hadn’t been for the fear that… infecting your loved ones, I would have loved it, because I wanted to be in the front line, I wanted to be there, I didn’t want to stay at home	P14, 60y
Q18	The consequences of facing ethical dilemmas	Physical, mental, and emotional health	I know… that it affected me, and what I lived was very hard, very hard, but I will still gradually realize the changes that it caused me. This is very traumatic for me	P1, 30y
Q19			I think that this was the worst experience of my life… I still feel bad	P15, 58y
Q20		Re-consider the profession	And well… you wonder if this profession… you wonder if it is worth it for you	P18, 38y
Q21			At the end of this process we came out strengthened, I mean, it was very positive for me, because I realized that I liked the critical patient. Indeed, I even considered enrolling in a Master’s programme. I liked it a lot	P12, 43y
Q22		Strengthened interpersonal relationships within teams	[...] helping each other. Yes, we bunched together a lot with our colleagues, no matter if they were doctors or assistants. We all worked hard because we all, I think, were living in such an extraordinary situation, we had to bunch together to be able to face it, right?	P1, 30y
Q23		Resilience	You have to get the aspects of learning, which help you to grow and be a better person and better professional. And that… well… I don’t know if that can happen again or if it will happen again. And you have to learn.	P20, 52y
Q24			As a professional I lived many things, many experiences, many of them bad but they make you grow. And as a person, the same, right? I lived things but I always try, at least I do, these to be positive for me.	P1, 30y
Q25		Key factors for a safe return	I understand that they are busy, a lot of things but… well… the rest of us do too and, in terms of twitching, this is not the moment, but we should have another meeting in which we talk about everything for real. What happened here? There hasn’t been a… I don’t know, a direct preoccupation. “How are you? How did it go… What… What do you need?” Basic things of common sense. And this didn’t… I didn’t have it anywhere. And now we are already back to normal activity	P7, 46y
Q26			Acknowledgement of what you’ve done, because I think that economically yes, but I don’t know… It’s like something is missing. Then you come back and it’s like you had come back here, and you work again and we said… hey, we need a bit of… right? Of rest.	P8, 31y
Q27			Especially training, training… And also information.	P13, 41y

## Data Availability

The datasets generated and/or analysed during the current study are not publicly available to guarantee anonymity but are available from the corresponding author upon reasonable request.

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
