# Peer review of "Moral Breakdowns and Ethical Dilemmas of Perioperative Nurses during COVID-19: COREQ-Compliant Study"

_healthcare, 2023, doi:10.3390/healthcare11131937_

Round 1
Reviewer 1 Report
REVIEW REPORT
Title: Moral breakdowns and ethical dilemmas of perioperative 2 nurses during COVID-19: COREQ-compliant study
Journal: Healthcare
Date completed: 25th June 2023.
Publisher: MDPI
General comments:
The study provides valuable insights into perioperative nurses’ experiences during the first wave of the COVID-19 pandemic. By exploring their personal and professional lives, the study sheds light on the specific challenges faced by these nurses in their daily work and how these challenges impact their overall well-being. The study focuses on moral breakdowns and ethical dilemmas triggered by the complex care situations that emerged during the pandemic. This analysis is crucial in identifying the specific ethical challenges faced by perioperative nurses and understanding the factors contributing to moral breakdowns in critical settings. The study identifies the context in which moral breakdowns emerged, providing a deeper understanding of the environmental and situational factors that contribute to these challenges. Such insights can help healthcare organisations and policymakers develop strategies to mitigate these factors and create a supportive environment for nurses. By exploring the consequences of facing ethical dilemmas, the study highlights the negative impact on professional responsibility and well-being among perioperative nurses. Recognising these consequences is essential for addressing the needs of nurses and implementing measures to support their mental health and job satisfaction.
The study adopts a qualitative design guided by a hermeneutical approach, allowing for an in-depth exploration of the nurses' experiences and perspectives. This methodology provides rich data and allows for a comprehensive understanding of the complex issues surrounding moral breakdowns and ethical dilemmas. The study concludes by emphasising the need for future research to identify ethical dilemmas during critical periods, develop strategies to enhance collaboration among colleagues and provide comprehensive support. These implications can guide the development of interventions, policies, and training programs to address ethical challenges and promote perioperative nurses' well-being. Overall, this study's findings have practical implications for healthcare organisations, policymakers, and researchers, providing insights into the experiences, challenges, and consequences faced by perioperative nurses during the COVID-19 pandemic. Addressing these issues makes it possible to develop strategies that support nurses' well-being and enable them to provide high-quality care in challenging situations.
The study is well-written, with its strengths in the methods and results sections. However, the work will benefit from some minor revisions. See the following comments and the ones made in the main manuscript PDF file.
1. Authors need to proofread their work for some grammatical errors.
2. Authors need to review existing literature showcasing what others have done and identifying key gaps.
3. If available, authors should provide the ethical approval number.
4. Authors should provide DOIs or web addresses of all electronic sources.

There are issues with punctuations, incomplete sentences, poor grammar. Only a few of these were detected. Authors should read through their work thoroughly for these. I have pointed some out in the attached PDF
Author Response
Thank you very much for your highly relevant comments to our manuscript. Please find below our point-by-point response to the reviewer comments which we hope will meet the requirements for acceptance. All suggestions have been implemented and discussed. All revisions have been formatted with track-changes in the revised manuscript.
Please see the attachment.

Reviewer 2 Report
This manuscript addresses the stress and ethical dilemmas and moral strain experienced by nurses during the height of COVID. It relies on qualitative interviews that explore the nature of the dilemmas nurses felt, focusing on the contextual factors, ethical dilemmas themselves, and the consequences that followed. The qualitative coding appears to have been done well, resulting in subcategories that add texture and nuance to the three larger categories.
The manuscript presents a solid rationale for the need for the study, as well as a theoretical framework. The literature review is complete without being excessively lengthy. Methodologically, the research has been conducted well, with a description of methods that add credibility to the study. The results are particularly interesting. Presenting the results in tabular form helps make clear to readers the nature of the ethical dilemmas, and the significant impact they had on the interviewees.
The writing is highly readable and engaging, well-organized and clear. There is one aspect that could be clarified. In Table 3 it is not clear to me why most subcategories have 2 examples, but the Ethical Dilemmas with Peers and Supervisors has many more. They are interesting, but a word or two about the reasoning for this would be helpful. (Perhaps they were more diverse, or greater in number? More significant to the informants or to the researchers?)
Finally, two small items:
For readability, avoid beginning successive sentences with the "however" (p. 1).
Table 2- a 'd' is missing in the word 'and'
In summary, this is a well-conceived and informative article. It adds to our understanding of the ethical dilemmas healthcare workers experienced during the recent pandemic.
Author Response
Thank you for your feedback. We appreciate your perspective, and as you mentioned, we agree that these examples were more significant to the informants due to their relevance in the work environment. We have taken your suggestion into consideration and have added this explanation to provide further clarity in the manuscript.
Please see the attachment

Reviewer 3 Report
First of all, I want to congratulate the authors of this study. In fact, it is a relevant topic for the academic and scientific community.
The manuscript has a good structure, and according the metodological criterion.
Only small observations:
line 43 - introduce final point.
It would be important to identify more limitations of this study.
Thank you.
Author Response
Thank you very much for your highly relevant comments to our manuscript. Please find below our point-by-point response to the reviewer comments which we hope will meet the requirements for acceptance. All suggestions have been implemented and discussed. All revisions have been formatted with track-changes in the revised manuscript.
Please see the attachment

Reviewer 4 Report
Congratulations.
The manuscript shows a study of interest, with a properly developed methodology and well-presented results that are useful for clinical nursing care.
Some minor comments to improve the manuscript are:
Introduction
I recommend that the authors include a nursing model, in addition to the Zigon's theoretical framework.
Method
Are the three researchers who coded the transcripts part of the authors (line 132 “All authors have PhD and have experience in qualitative research”)? What profile do they have to perform this function? I request the authors to review this aspect to better clarify it in the manuscript.
Ethics
The section is correct, and the authors can include the approval code of the Ethics Committee document attached (IIBSP-COV-2020-58)
Discussion
Line 245 “This study is the first of its kind to examine the morality of the nursing profession within the context of a pandemic, specifically from the perspective of nurses who rapidly adapted to new roles, tasks, and activities.” A similar sentence appears on line 310
The study may be original and not find similar studies carried out during the pandemic months, but there are other studies on coping and the moral and ethical area of nursing professionals in relation to COVID-19. Even if it is an analysis after the months of pandemic. Authors must modify the statement for other alternatives such as “The study is one of the first…”, “The study is one of the few…”. In addition, the authors can include in the discussion a reflection on whether there would be differences in the results when doing during the period or later.
A similar study can be:
Jia Y, Chen O, Xiao Z, Xiao J, Bian J, Jia H. Nurses' ethical challenges caring for people with COVID-19: A qualitative study. Nurs Ethics. 2021 Feb;28(1):33-45. doi: 10.1177/0969733020944453. Epub 2020 Aug 28. PMID: 32856534; PMCID: PMC7653013.
Moreover, another similar study (already cited in the manuscript) is:
Hossain F, Clatty A. Self-care strategies in response to nurses' moral injury during COVID-19 pandemic. Nurs Ethics. 2021 Feb;28(1):23-32. doi: 10.1177/0969733020961825. Epub 2020 Oct 30. PMID: 33124492; PMCID: PMC7604672.
Please modify these aspects and thus the manuscript will be approved.
Author Response

(The authors gave the same response as above.)
